

# Adaptation prevents the extinction of *Chlamydomonas reinhardtii* under toxic beryllium

Beatriz Baselga-Cervera[1], Eduardo Costas[1], Estéfano Bustillo-Avendaño[1,2] and Camino García-Balboa[1,2]

[1] Genetics, Department of Animal Production, Universidad Complutense de Madrid, Madrid, Spain
[2] I+D+I, Biotechnological Environmental Solutions S.L., Madrid, Spain

Corresponding author
Camino García-Balboa,
camino@ucm.es

## ABSTRACT

The current biodiversity crisis represents a historic challenge for natural communities: the environmental rate of change exceeds the population's adaptation capability. Integrating both ecological and evolutionary responses is necessary to make reliable predictions regarding the loss of biodiversity. The race against extinction from an eco-evolutionary perspective is gaining importance in ecological risk assessment. Here, we performed a classical study of population dynamics—a fluctuation analysis—and evaluated the results from an adaption perspective. Fluctuation analysis, widely used with microorganisms, is an effective empirical procedure to study adaptation under strong selective pressure because it incorporates the factors that influence demographic, genetic and environmental changes. The adaptation of phytoplankton to beryllium (Be) is of interest because human activities are increasing the concentration of Be in freshwater reserves; therefore, predicting the effects of human-induced pollutants is necessary for proper risk assessment. The fluctuation analysis was performed with phytoplankton, specifically, the freshwater microalgae *Chlamydomonas reinhardtii*, under acute Be exposure. High doses of Be led to massive microalgae death; however, by conducting a fluctuation analysis experiment, we found that *C. reinhardtii* was able to adapt to 33 mg/l of Be due to pre-existing genetic variability. The rescuing adapting genotype presented a mutation rate of $9.61 \times 10^{-6}$ and a frequency of 10.42 resistant cells per million wild-type cells. The genetic adaptation pathway that was experimentally obtained agreed with the theoretical models of evolutionary rescue (ER). Furthermore, the rescuing genotype presented phenotypic and physiologic differences from the wild-type genotype, was 25% smaller than the Be-resistant genotype and presented a lower fitness and quantum yield performance. The abrupt distinctions between the wild-type and the Be-resistant genotype suggest a pleiotropic effect mediated by an advantageous mutation; however, no sequencing confirmation was performed.

## INTRODUCTION

A major concern worldwide is the present status of the world's freshwater reserves. Human activities are changing not only the physical-chemical characteristics of these

water environments but also their species and population distributions (*Balmford, Green & Jenkins, 2003*; *Smith & Bernatchez, 2008*). These events are not isolated incidents; the impact of human actions has increased so markedly during recent years that humans may have become the world's dominant selective force (*Hendry & Kinnison, 1999*; *Palumbi, 2001*; *Stockwell, Hendry & Kinnison, 2003*). These changes are dramatically accelerating shifts in the number and demography of species and leading to a biotic turnover of the world's biosphere and a loss of habitats and biodiversity (*Kolbert, 2014*). Understanding both the ecological and genetic responses of populations under scenarios in which adaptation over short time scales might preclude population extinction is critical to identifying pathways and features that enable the recovery of declining populations and the preservation of species.

Microorganisms, particularly extremophile organisms, have long been known to present a substantial ability to tolerate and rapidly adapt to extreme environmental stress conditions (*López-Rodas et al., 2009*; *Huertas et al., 2010*). Recently, microbes have gained significance in the biodiversity crisis, as some ecological cycles will collapse if microorganisms succumb to pollutants (*Woodruff, 2001*). One of the major groups of aquatic organisms—part of the unseen majority—is phytoplankton. Phytoplankton are the principal primary producer in water environments (*Falkowski & Raven, 2013*) and are likely the main oxygen producer on a global scale. Since 1950, the amount and productivity of phytoplankton on the earth have exhibited decreasing trends (*Behrenfeld et al., 2006*; *Boyce, Lewis & Worm, 2010*). Disturbingly, these decreasing rates appear to be related to anthropogenic stressors, and some contaminants significantly affect phytoplankton (*García-Villada et al., 2004*; *Stauber et al., 2008*; *Sánchez-Fortún et al., 2009*; *Marvá et al., 2014a*).

The selection pressure of anthropogenic stressors typically involves a demographic cost in which a reduction in population size occurs and changes the population's fitness. Selective pressures that are fatal for most of the population push the diverse array of adaptation pathways to their limits depending on the presence of rescue phenotypes that are able to withstand these pressures (*Gonzalez et al., 2013*). Most eco-toxicity studies have addressed the immediate effects of pollutants upon populations without considering the evolutionary, long-term and trans-generational effects of the adaptation process (*Bickham, 2011*; *Huertas et al., 2011*). Understanding both the ecological and genetic responses of populations under scenarios in which adaptation over short time scales might occur is critical to identifying favorable conditions that enable the recovery of declining populations (i.e., in the context of evolutionary rescue (ER)). A difficulty in studying adaptation by means of eco-toxicological evolutionary approaches arises from the temporal misalignment between exposure and response as many factors might affect the response during the interim (*Pomati & Nizzetto, 2013*). Notwithstanding the challenges of extrapolating data, trans-generational and eco-evolutionary studies of pollutant effects on phytoplankton are essential to forming reliable predictions about the future biosphere.

The hazards of contaminants in the aquatic framework are a critical component of ecological risk assessment (ERA). Recently, a variety of approaches have been proposed that aim to reduce existing uncertainties in risk assessment knowledge by means of information integration, such as integrating testing strategies (ITS) or evolutionary toxicity (*WHO,*

*2001*; *Bradbury, Feijtel & Leeuwen, 2004*). Under this integration perspective, laboratory experiments of phytoplankton that address ecological and evolutionary issues are necessary (*Myers et al., 2000*), while acknowledging the limitations of extrapolating the results to field conditions. Such studies can provide an improved understanding of toxicity across different layers of biological organization (*Brooks, Fulton & Hanson, 2015*) and increase the reliability of ERA.

An example of a human activity that exerts strong selection and thereby accelerates evolutionary change is extreme metal pollution of waters, such as that of mining ponds or mining spills (*Baos et al., 2002*; *Costas et al., 2007*; *López-Rodas et al., 2008*; *García-Balboa et al., 2013*). One compound involved in such pollution is beryllium (Be), an atypical metal with unusual proprieties: it is one of the lightest structural metals and presents high water solubility due to its highly electropositive divalent base cation. It has no known biological function; furthermore, it is one of the most toxic elements to both humans and the aquatic environment (*Edmunds & Trafford, 1993*; *Vesely et al., 2002*). At the cellular level, $Be^{2+}$ ions compete with other ions of the same charge, such as $Mg^{2+}$, to bind to phosphate groups of nucleotides and nucleic acids. This action may be involved in the inhibitory effect of beryllium over alkaline phosphatase, which plays a key role in dephosphorilation processes in some organs and tissues. Furthermore, beryllium salts have been shown to induce chromosomal aberrations and chromatid exchanges in mammalian cells but to have no mutagenic effects in bacterial assays (*Vesely et al., 2002*; *Gordon & Bowser, 2003*). Small quantities of Be are naturally present in the environment; however, human activities release beryllium to surface waters though release runoff from beryllium-containing waste sites and via the effluents of treated wastewater from beryllium and related industries, such as the nuclear industry and copper-beryllium alloy production (*US Environmental Protection Agency, 1980*; *US Environmental Protection Agency, 1981*). According to the Toxics Release Inventory (TRI), a total of 33,647 kg of beryllium and 412,990 kg of beryllium compounds were released into the environment in 1999 (*TRI99, 2002*). Disturbingly, Be mobilization increases with acidification (*Neal, 2003*), and one study found surface natural water concentrations of up to $99.3 \pm 42.1$ µg/l in 6.1% of samples (*US Environmental Protection Agency, 2001*). Additionally, concentrations of the cosmogenic natural radionuclide $^7$Be in marine ambiance and bioaccumulation in marine organisms, such as brown algae (*Marsh & Buddemeier, 1984*; *Ishikawa, Kagaya & Saga, 2004*), have been reported.

Laboratory experiments constitute a potential "halfway path" between standardized protocols and field experiments. We propose evolutionary, trans-generational experiments of phytoplankton adaptation to anthropogenic pollutants as appropriate tools for obtaining information that is necessary for suitable ERA in the aquatic ambiance. In this study, the population evolutionary dynamics and morphological variation of the species *Chlamydomonas reinhardtii* in the presence of dissolved Be was evaluated to determine the tolerance level of the wild type, with massive microalgae death occurring at 33 mg/l. Then, the microalgae was exposed to a selective pressure of Be at 33 mg/l in a fluctuation experiment to determine its ability to adapt by assessing the physiological and behavioral changes in the population and the distinctive characteristics of the resistant population. *Chlamydomonas reinhardtii* was able to adapt by means of pre-selective

adaptation; i.e., by achieving Be resistance prior to exposition. Moreover, the resistant strain obtained presented significant physiological and morphological differences from the wild type.

## MATERIAL AND METHODS

### Phytoplankton organisms and culture conditions

*Chlamydomonas reinhardtii* Dangeard (wild-type strain Chla) wild-type mesophilic Chlorophyceae was obtained from the Universidad Complutense de Madrid algae culture collection. This strain was isolated from a freshwater pond in Doñana National Park, Andalucía, Spain. The isolation site was not polluted by any detectable metal or radiation.

The strain was cultured axenically in 20 mL of BG-11 medium (Sigma–Aldrich Chemie, Taufkirchen, Germany) into 100 mL cell culture flasks (Greiner, Bio-One Inc., Longwood, NJ). The temperature and light conditions were 22 °C and 80 mol m$^{-2}$ s$^{-1}$ over the wavelength (ranging from 400 to 700 nm) under continuous illumination.

The clonal strain was maintained in a mid-log exponential growth by serial transfers of 100 cells every fifteen days to fresh BG-11 medium.

### Toxic effect of beryllium on *C. reinhardtii*

The toxicity of beryllium was evaluated with a dose–effect relationship. The reagent used for the experiment was a beryllium sulfate tetrahydrate (BeSO$_4$ • 4H$_2$O) with 98% purity level (Alfa Aesar, Haverhill, MA, USA). The beryllium salt was diluted in BG-11 filtered medium and subsequently re-diluted to obtain concentrations of 0.8, 4, 8, 16, 33 and 66 mg/l. For each dose and the unexposed control, three replicates of the Chla wild-type strain in the exponential growth rate phase were inoculated with $1.5 \times 10^5$ cells/mL. The relationship between growth rate (m) and different dosage levels up to 28 days was analyzed. The number of cells was counted in Uriglass settler chambers (Fast read; Biosigma, Venice, Italy) with an inverted microscope with a coupled fluorescence lamp (Axiovert 35; Zeiss, Oberkochen, Germany). Both short-term (up to 96 h) and long-term (up to 28 days) actions were evaluated to assess the acute and multi-generational toxicity. Multi-generational effect are the changes across several generation of exposure to Be.

### Adaptation capability of *C. reinhardtii* to beryllium

A modified fluctuation analysis for liquid media (*López-Rodas et al., 2001*) was performed based on the pioneering approach of *Luria & Delbrück (1943)*. This experiment evaluated the capability of microorganisms (in our case, *C. reinhardtii*) to resist to a high dose of a stressor (beryllium) after a massive die-off of sensitive cells. Moreover, it allows for differentiation between evolutionary forces: if the resistance to beryllium results from the appearance of a spontaneous mutation prior to exposure (pre-selective adaptation) or from the adaptation of the strain caused by an environmental stress (post-selective adaptation) by acclimatization or post-adaptive mutations.

A schematic diagram of the modified fluctuation analysis is shown in Fig. 1. Two different sets of experiments, one acting as treatments (Set 1) and the other as controls (Set 2), were performed in gas exchange plastic tubes with BG-11 medium. All the cells

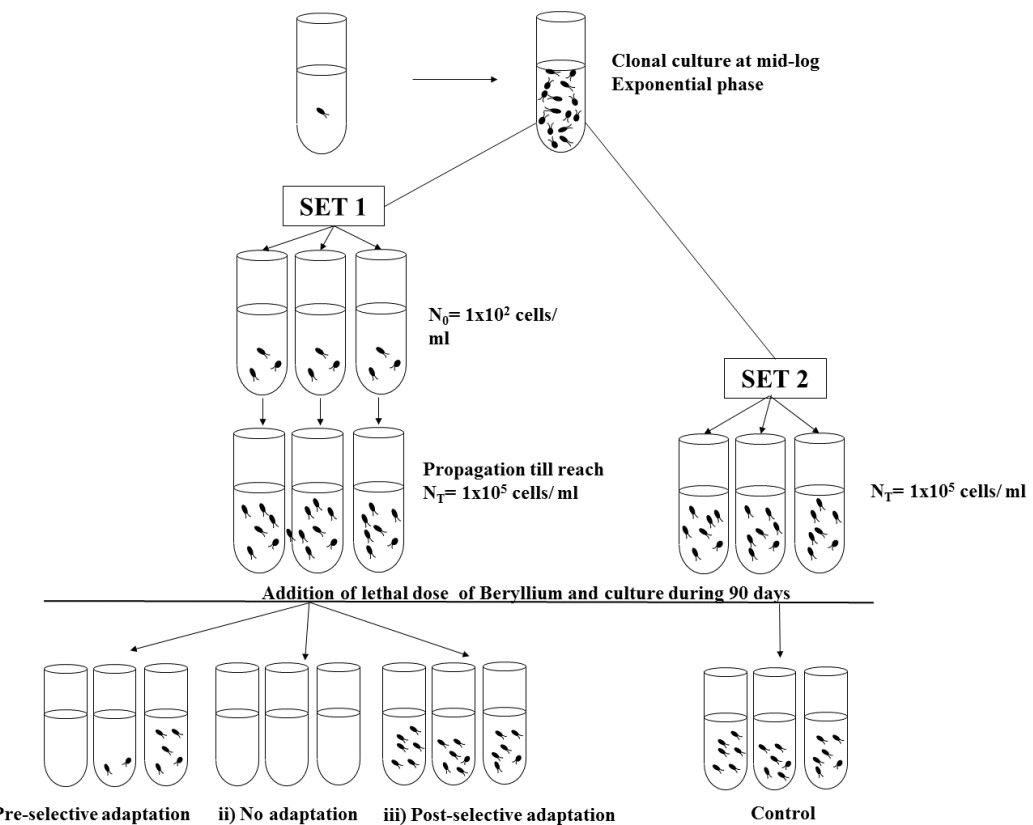

i) Pre-selective adaptation    ii) No adaptation    iii) Post-selective adaptation    Control

**Figure 1  Schematic representation of the modified fluctuation test.** Two different sets, Set 1 and Set 2, were established with cells from the same parental population. First, Set 1 cultures were inoculated with an initial inoculum ($N_0$) and propagated under non-selective conditions till reach $N_t$ cells density. Once Set 1 cultures reached $N_t$ cells, Set 2 cultures (control cultures) were founded with an initial cells inoculum of the same density as $N_t$. Thereafter, both Sets were exposed to a lethal dose of Beryllium and maintained under selective conditions during 90 days. Three mutually exclusive possible scenarios could result from the Set 1 trials: (i) pre-selective adaptation, (ii) no adaptation and (iii) post-selective adaptation.

inoculums were from the same parental population isolated from a single cell (clonal propagation), to prevent spontaneous mutation accumulation.

First, in Set 1 (treatment trials), 95 replicate tubes were inoculated with a low population density ($N_0 = 10^2$ cells), which ensured the absence of pre-existing mutants. The number of replicate tubes was calculated prior to the experiment to obtain suitable precision according to *Li et al. (1983)*. These replicates were allowed to propagate until a high population density was reached ($N_t = 10^5$ cells); during the cell division process (which spanned an average of 11.68 generations), several random mutations affecting the sensitivity to resistance arose at different times. At the end of the propagation process, fresh medium with the selective agent (33 mg/l of beryllium, which was the no-viability dose identified in the dose–effect assays) was added to all of the replicates.

For Set 2 (control trials), at the end of the growing phase of Set 1, 47 replicates (half of the parallel tubes in Set 1) with $N_0 = 10^5$ cells from the same parental population as Set 1 and with selective medium containing 33 mg/l of Be were established.

Both sets of trials were maintained under the same culture conditions at 22 °C with continuous light for 90 days. After this period, all of the tubes were blindly counted (thereby ensuring that one resistant cell could propagate to a sufficient extent as to be detected).

From the fluctuation analysis, according to *Luria & Delbrück (1943)*, three mutually exclusive possible outcomes could result:

(i) Post-selective adaptation: If resistant variants arise by acclimatization or post-adaptive mutation, each cell presents the same probability of adapting in the selective medium such that the number of resistant cells should be similar among all of the replicates. Therefore, the inter-culture variation will follow a Poisson model (a variance/mean ratio similar to 1 and similar between Sets 1 and 2), presenting low variation in cell number.

(ii) Pre-selective adaptation: If resistant variants arise prior to exposure via random mutations that emerge spontaneously in the population, each parallel tube presents a unique probability per cell division of generating a resistant variant. Some parallel cultures will achieve resistant variants earlier than others during the propagation and will propagate resistant progeny. The tube-to-tube variation will be high in Set 1 and will not follow a Poisson model (a variance/mean ratio higher than 1, with a significant difference between Set 1 and Set 2). Set 2, acting as a control, will present low inter-culture variation. Consequently, if resistant variants arise by *a priori* mutations, then some replicates in Set 1 should show no resistant cells and others show high densities of resistant cells, whereas in Set 2, all of the replicates should show resistant variants.

(iii) No adaptation: If no resistant variants arise, then no living cells should be detected in any of the replicates of Set 1 or Set 2.

Fluctuation analysis also enables the estimation of the mutation rate ($\mu$), i.e., the rate of appearance of resistant cells to the selective agent. From the proportion of replicates that show no resistant cells ($P_0$), we can estimate $\mu$ according to the following equation:

$$\mu = \mathrm{Log}_e P_0 / (N_t - N_0) \text{ (\textit{Luria \& Delbrück, 1943})}$$

where $\mathrm{Log}_e$ is the natural algorithm, $P_0$ is the proportion of replicates in Set 1 that show no resistant cells at the end of the experiment, $N_0$ is the cell number at the beginning of the experiment and $N_t$ is the number of cells at the end of the experiment.

## Characterization of the beryllium-resistant population

Population characteristics, such as the growth rate ($m$) or coefficient of selection ($s$) of a population, facilitate an improved understanding of population dynamics. The Be-resistant variants from five different parallel tubes from Set 1 were randomly chosen and maintained under selective conditions for 60 days to establish a Be-resistant *Chlamydomonas* sp. strain. Subsequently, a small population of the resistant strain was cultured in fresh medium without Be and characterized under non-selective conditions.

The growth rate (Malthusian component of fitness ($m$)) is the rate of change of the number of individuals in a population in a time interval, and it was calculated with the following equation:

$$N_t = N_0 e^{mt} \text{ (\textit{Crow \& Kimura, 1970})}$$

where $t = 7$ days and $N_0$ and $N_t$ represent cell density at the beginning and end of the experiment.

The coefficient of selection or the selective elimination rate ($s$) results from the comparison of two alleles of different fitnesses and is an estimate of the relative fitness of the less-favored allele. Phytoplankton s values calculated in natural populations range between 0.03 and 0.1 depending on species (*Fox, Nelson & Mccauley, 2010*) and between 0.46 and 0.84 under selective conditions (*López-Rodas, Maneiro & Costas, 2006*). It is calculated by the following equation:

$$s = 1 - (m^r - m^w)$$

where $m^r$ and $m^w$ are the Malthusian fitnesses under non-selective pressure of the beryllium-resistant population and the wild-type parental population.

When the mutation from wild-type cell to resistant variant occurs repeatedly in the population, new mutants will appear each generation. If the fitness of the mutant allele is lower than the wild-type allele, the resistant alleles will be eliminated from the population by chance or natural selection; however, the population will always have a number of mutant alleles that persevere. In these cases, the equilibrium of mutation-selection ($q$) indicates the number of resistant mutants as a function of the selective pressure present in the population and can be estimated as the balance between $\mu$ and $s$ according to the following equation by *Kimura & Maruyama (1966)*:

$$q = \mu/(\mu + s).$$

Phytoplankton q values under selective conditions range between 1.2 resistant cells (*Marvá et al., 2014a*) and 6,500 resistant cells (*López-Rodas, Maneiro & Costas, 2006*) per million cells in the population.

### Transmission electron microscopy (TEM)

To assess the possible intracellular morphological differences between *Chlamydomonas* sp. Be-resistant cells and the wild-type parental cells, we performed a TEM analysis. Before the fixation of the two samples, the wild-type cells in BG11 medium and the Be-resistant cells in BG11 medium with 33 mg/l of beryllium were rinsed three times with a saline phosphate buffer by centrifugation at 4000 rpm for 5 min. The resulting pellet was preserved in EM fixative (2.5% glutaraldehyde in sodium cacodylate-sacharose buffer, pH 7.2, at 4 °C for 24 h) and post-fixed in a 2% osmium tetroxide ($OsO_4$) buffer for one hour at 4 °C, dehydrated in an increasing series of acetone, and embedded in Spurr resin. Afterwards, the samples were sectioned in 80-nm-thick sections with a LKB 2088 ultramicrotome and collected on copper grids. All of the reagents were purchased from Sigma-Aldrich Chemical Company (St. Louis, MO, USA).

The TEM images were obtained at 100 kV with an electron microscope, JEOL JEM-2010 (Jeol Ltd., Tokyo, Japan).

### Morphometric and stereological variation

Morphometric analysis was conducted to quantitatively characterize cell ultrastructural organization and size. By means of stereology, we performed a morphometric analysis

comparing the ultrastructure and cell volumes of the wild type and resistant variants of *Chlamydomonas* sp.

TEM ultrastructure micrographs were fabricated from each ultrathin section at 12000x. The number of micrographs of each organelle was estimated with Williams's progressive mean technique (*Williams, 1977*) with a ±5% confidence limit, corresponding to approximately 25 micrographs in each case. The stereological and morphometric methodologies have been previously described in detail (*Costas et al., 1988a*; *Costas et al., 1988b*; *Costas et al., 1992*). The procedures used to calculate the micrograph profile areas were the point-counting procedure and planimetry. (*Weibel & Bolender, 1973*) (a schematic representation of the point-counting procedure is provided in Fig. S1).

The morphometric parameters studied were as follows:

1. Absolute volumes ($V_x$):

$$V_x = \frac{Vv_x}{Nv_x}$$

   where $Vv_x$ is the relative volume of the component $x$ and $Nv_x$ is the numerical density of the component $x$.

2. $Vv_x$ = volume density or relative volume of the component $x$, which indicates the relative volume of the component $x$ with respect to the total cell volume. The volume density $x$ was calculated as the relationship between the sectional areas and the reference in the micrograph ratio according to the principle of Delesse (*Weibel & Bolender, 1973*).

3. $Nv_x$ = numerical density of the component $x$ or the relative numerical density of the $x$ component, which indicates the number of components of $x$ per $\mu m^3$. It is calculated as

$$Nv_x = \frac{(KNa^{3/2})}{(\beta Vv_x)}$$

   where $Na$ is the numerical profile density and $K$ and $\beta$ are the coefficients of form, which estimate the diameter of dispersion ($K$) and the mean axial ratio ($\beta$) of the profile population.

   The coefficient of form $\beta$ was calculated by using a normogram after determining the mean axial ratio. (*Weibel & Bolender, 1973*).

$$K = \left(\frac{M_3}{M_1}\right)^{\frac{3}{2}}$$

   where $M_3$ and $M_1$ are the mean diameter of each profile.

4. The coefficient of form of the diameters of the axis (C.F.d):

$$C.F.d = \frac{d}{D}$$

   where $d$ is the transverse axis and $D$ is the longitudinal axis.

5. The coefficient of form for circular profiles (C.F.c):

$$C.F.c = \frac{4\pi\,\text{Area}}{(\text{Perimeter})^2}.$$

For the stereological study, the Chla wild-type and Chla Be-resistant cells were divided into five cellular compartments: nucleus (n), chloroplast (chl), mitochondria (m), storage products (s) (including vacuoles, inclusions and droplets) and total cells (c).

## Imaging-PAM fluorescence valuation

PAM fluorometers emit pulse-modulated light to excite the fluorescence of chlorophyll *a* to evaluate the photosynthetic performance of algae and plants.

A Maxi-Imaging-PAM chlorophyll a fluorometer (Heinz Walz GmbH, Effeltrich, Germany) was used to measure the chlorophyll fluorescence of *Chlamydomonas* sp. This device is a high-resolution fluorometer that measures sample areas (selected areas of interest (AOI)), allowing the assessment of algae suspensions in multiwell plates. The Imaging-PAM analyzes the images of fluorescence emission captured by a couple device LED array and a filter-covered CCD camera synchronized with the light emission (the Imaging-PAM specifications are described in detail in *Ralph et al. (2005)*.

To quantify the fluorescence values, we blindly counted the cells of the *Chlamydomonas* sp. wild-type strain and the Be-resistant strain with Uriglass settler chambers. In 48-well plates, we filled 15 wells of each strain with $5 \times 10^6$ cells. Three wells of each strain were exposed to concentrations of 4, 8, 16, 33 mg/l of Be, and three unexposed replicates of each strain were used as controls. Measurements were taken the first 16 h and at 24, 48 and 72 h after exposure. We adapted the cells in the multiwell plate to darkness 30 min before the measurements.

Then, images of the dark-adapted state were recorded, with an image of the dark-adapted minimum fluorescence state ($F_0$) before the application of an excitation pulse (2700 μE $m^{-2}$ $s^{-1}$ provided by the blue LED ring at 470 nm) providing the dark-adapted maximum fluorescence state (Fm). Subsequently, light-adapted images were induced during a cycle of 10 measurements by the application of a saturation pulse (SP) (2700 μE $m^{-2}$ $s^{-1}$ blue LED light for 800 ms). During the cycle, the current fluorescence yield (*Ft*) in the AOI with a very low actinic light (2 Hz modulated blue excitation light of 470 nm and intensity of 0.5 μE $m^{-2}$ $s^{-1}$) was continuously measured to ensure minimum fluorescence (steady state). Images of the light-adapted estate were recorded immediately before the SP (the current fluorescence (*Ft*)) and at the end of the SP (quantifying the value of maximal fluorescence (*F′m*)). From the *F′m* and *Ft* values, the effective PS II quantum yield *(Y(II))* was calculated (*Genty, Briantais & Baker, 1989*):

$$Y(II) = (F'm - Ft)/F'm$$

Phytotoxicity was quantified with the values of $Y(II)$ of the control and the study samples as the relative inhibition of $Y(II)$:

$$Inh.(\%) = \big[(Y(II)\ control - Y(II)\ sample)/Y(II)\ control\big] \times 100\%.$$

Values above 10% of inhibition of the $Y(II)$ with regard to the control were considered phytotoxic.

The recorded digital images were analyzed with analytical software (Imaging-WIN; Walz, Effeltrich, Germany), and the numerical values of the fluorescence parameters of chlorophyll *a* were obtained.

## Statistical analysis

The non-parametric Mann–Whitney $U$ test (two-tailed) was performed to detect the significance of ultrastructural differences between wild-type sensitive *Chlamydomonas* sp. cells and Be-resistant mutant *Chlamydomonas* sp. cells. All of the statistical analyses were performed according to *Siegel (1956)*. A $X^2$ goodness-of-fit test at $p < 0.01$ was performed to compare the differences in the variance/mean ratio between Set 1 and Set 2.

## RESULTS

### Toxic effect of beryllium to *Chlamydomonas* reinhardtii

Beryllium was found to be toxic to *Chlamydomonas* sp. (viability is shown in Fig. S2). In less than 24 h at the lowest dose of 0.8 mg/l, the viability decreased to less than half of the control viability. Moreover, no detectable living cells at 24 h were detected in the replicates with 33 and 66 mg/l Be, and no viability was observed at 28 days (viability was directly measured by counting fluorescence-emitting cells with an inverted microscope with a coupled fluorescence lamp). Based on these results, the dose of 33 mg/l Be was selected for the fluctuation analysis.

Furthermore, we obtained striking results at doses below 33 mg/l: following one week of exposure, the viability differences between the control and the exposed populations decreased. This phenomenon was most pronounced at 16 mg/l on the 14th day, as no living cells were observed during the first week at this dose, but during the second week, the population began to recover. This recovery phenomenon at the lower doses may be explained by the plasticity of the population and, specifically at 16 mg/l, by the propagation of the resistant cells, which might rescue the population from extirpation.

### Adaptation capability of *Chlamydomonas* sp. to beryllium

Our first aim was to determinate if *Chlamydomonas* sp. was capable of adapting to 33 mg/l of Be and, if so, to determinate the nature of the adaptation (pre-selective adaptation, by means of existing mutations prior to the exposure, or post-selective, by means of plasticity or *novo* mutations).

Although *Chlamydomonas* sp. was apparently unable to survive at 33 mg/l (ostensibly undergoing a population die-off) after one month of exposure, the fluctuation analysis yielded a different outcome. At the beginning of the exposure, massive cell die-offs occurred in both Set 1 and Set 2. However, after 90 days of incubation under the same conditions, several living cells persisted in different tubes from Set 1 and in all of the tubes from Set 2 due to the proliferation of beryllium-resistant variants (Table 1). Variation in the resistant cell density between parallel replicates in Set 1 (from 0 to over $10^6$) was observed, indicating the presence of fluctuation (variance/mean ratio > 1 ($p < 0.01$ according to a $X^2$ goodness-of-fit test)). The strong fluctuation present in Set 1 indicates that Be-resistant variants rose prior to exposition by means of pre-selective adaptation. In Set 2, i.e., the

**Table 1 Fluctuation analysis of the occurrence of Be resistant variants of *C. reinhardtii*.**

|  | Set 1 | Set 2 |
|---|---|---|
| Number of replicate cultures | 87[a] | 47 |
| No. of cultures containing: |  |  |
| 0 resistant cells | 6 | 0 |
| From $1 \times 10^3$ to $2.5 \times 10^4$ total resistant cells | 46 | 47 |
| From $2.5 \times 10^4$ to $5 \times 10^5$ total resistant cells | 17 | 0 |
| From $5 \times 10^5$ to $1 \times 10^6$ total resistant cells | 10 | 0 |
| From $1 \times 10^6$ to $2 \times 10^6$ | 7 | 0 |
| More than $2 \times 10^6$ | 1 | 0 |
| Fluctuation | Yes | No |
| Adaptation mechanisms | Pre-selective adaptation | |
| Mutation rate ($\mu$) | $9.61 \times 10^{-6}$ | |

**Notes.**
[a] Eight of the initial parallel tubes of Set 1 could not be counted.

control trials, the number of cells did not show large variation between replicates (from $10^3$ to over $10^5$), indicating weak fluctuation compared with that of Set 1 (p < 0.01 according to a $X^2$ goodness-of-fit test). The low level of fluctuation in Set 2 indicates that the large level of inter-tube fluctuation in Set 1 must be the result of processes other than sampling error.

The mutation rate calculated *a posteriori* of beryllium-resistant *Chlamydomonas* was $9.61 \times 10^{-6}$ cells per division.

## Population dynamics of the beryllium-resistant organisms

The growth rate of beryllium mutants cultured in BG-11 medium without beryllium was lower than that of the wild-type strains used as parental populations. The Malthusian component of fitness was 0.4129 for the wild-type strain and 0.3344 for the beryllium-resistant strain, yielding a selection coefficient (s) of 0.9215. This *s* value is one of the lowest calculated under selective pressure (*López-Rodas, Maneiro & Costas, 2006*). The frequency (*q*) of beryllium-resistant cells in the population was calculated as the mutation-selection balance, resulting in 10.42 mutant cells per million of wild-type cells in the parental population cultured without beryllium selective pressure. This *q* value is low compared with the range of calculated balances (*López-Rodas, Maneiro & Costas, 2006*). This frequency result explains the generation of beryllium-resistant cells with each generation in large populations of the wild-type strain and their sustained elimination by natural selection.

## Morphometric and stereological variation: from macroscopic to TEM analysis

The first detectable phenotypical difference was observed macroscopically by comparing the behavior of the population in the culture flasks in liquid medium. Beryllium-resistant strains were distributed in aggregates at the bottom of the cell culture flasks, whereas the wild-type *Chlamydomonas* presented a uniform distribution in the culture medium.

**Table 2** Differences in cell shape and ultrastructural organization between wild-type and Be-resistant *Chlamydomonas* sp. (mean ± se) (se < ±0.10).

| Trait (units) | Wild-type *Chlamydomonas* sp. (Be-sensitive) | Be-resistant *Chlamydomonas* sp. (Mutant) | Mann–Whitney U test |
|---|---|---|---|
| $V_c$ ($\mu m^3$) | $242 \pm 39$ | $180 \pm 37$ | *($p < 0.001$) |
| C.F.d | $0.86 \pm 0.04$ | $0.98 \pm 0.05$ | *($p < 0.01$) |
| C.F.c | $0.79 \pm 0.06$ | $0.94 \pm 0.04$ | *($p < 0.01$) |
| $Vv_{n,c}$ ($\mu m^3$) | $0.20 \pm 0.07$ | $0.23 \pm 0.08$ | ns ($p > 0.5$) |
| $Vv_{chl,c}$ ($\mu m^3$) | $0.31 \pm 0.05$ | $0.37 \pm 0.07$ | *($p < 0.01$) |
| $Vv_{m,c}$ ($\mu m^3$) | $0.07 \pm 0.03$ | $0.11 \pm 0.06$ | *($p < 0.01$) |
| $Vv_{s,c}$ ($\mu m^3$) | $0.08 \pm 0.04$ | $0.03 \pm 0.02$ | *($p < 0.01$) |
| $Nv_{chl}$ ($\mu m^{-3}$) | $5.47 \times 10^{-3} \pm 0.13 \times 10^{-3}$ | $9.33 \times 10^{-3} \pm 0.51 \times 10^{-3}$ | *($p < 0.01$) |
| $Nv_m$ ($\mu m^{-3}$) | $0.080 \pm 0.004$ | $0.092 \pm 0.006$ | *($p < 0.01$) |

Notes.
*Statistically significant.
nucleus (n), chloroplast (chl), mitochondria (m), storage products (s), total cells (c).
$V_x$, absolute volume of component x; C.F.d, coefficient of form diameter; C.F.c, coefficient of circular form; $Vv_{x,r}$, volume density of component $x$ ($x$ being the component of study) related to component $r$ ($r$ being the total cell value); $Nv_x$, numerical density of component $x$.

*Chlamydomonas reinhardtii* is known for its ability to form palmelloid colonies as a response to environmental changes (*Iwasa & Murakami, 1968*; *Iwasa & Murakami, 1969*; *Lurling & Beekman, 2009*). The optic microscopy valuation suggested such behavior (Fig. S3). According to the optic microscopic valuation, *Chlamydomonas* sp. Be-resistant cells presented a smaller size than did the cells of the parental population (Fig. 2) and did not show any active motion, suggesting that the flagellum was possibly damaged. The TEM images consistently showed an increased presence of starch granules within the chloroplasts (Fig. 2), which is related to a change in carbon storage characteristics that occurs in response to stress.

The stereological analysis of cell absolute and relative volume and organelle densities are summarized in Table 2. The table quantitatively characterizes the main ultrastructural features of wild-type and Be-resistant *Chlamydomonas* sp. The strains showed differences in ultrastructural organization. The absolute volumes of the Be-resistant *Chlamydomonas* sp. cells were significantly lower than those of the wild-type strain at $p < 0.001$, supporting the qualitative differences evaluated by optic microscopy. The relative volumes of the chloroplast and mitochondria were higher in the Be-resistant *Chlamydomonas* sp. strain, whereas the storage product's relative volume was lower relative to the wild-type strain ($p < 0.01$). There was also a significant increase ($p < 0.01$) in the numerical densities of the chloroplasts and mitochondria of the Be-resistant *Chlamydomonas* sp. strain. No significant differences were found between the two strains in the relative volume of the nucleus.

### Imaging-PAM fluorescence

The *Y(II)* of both wild-type and Be-resistant *Chlamydomonas* sp. strains was measured in triplicate at different Be doses for up to 72 h. The *Y(II)* of the unexposed wild-type

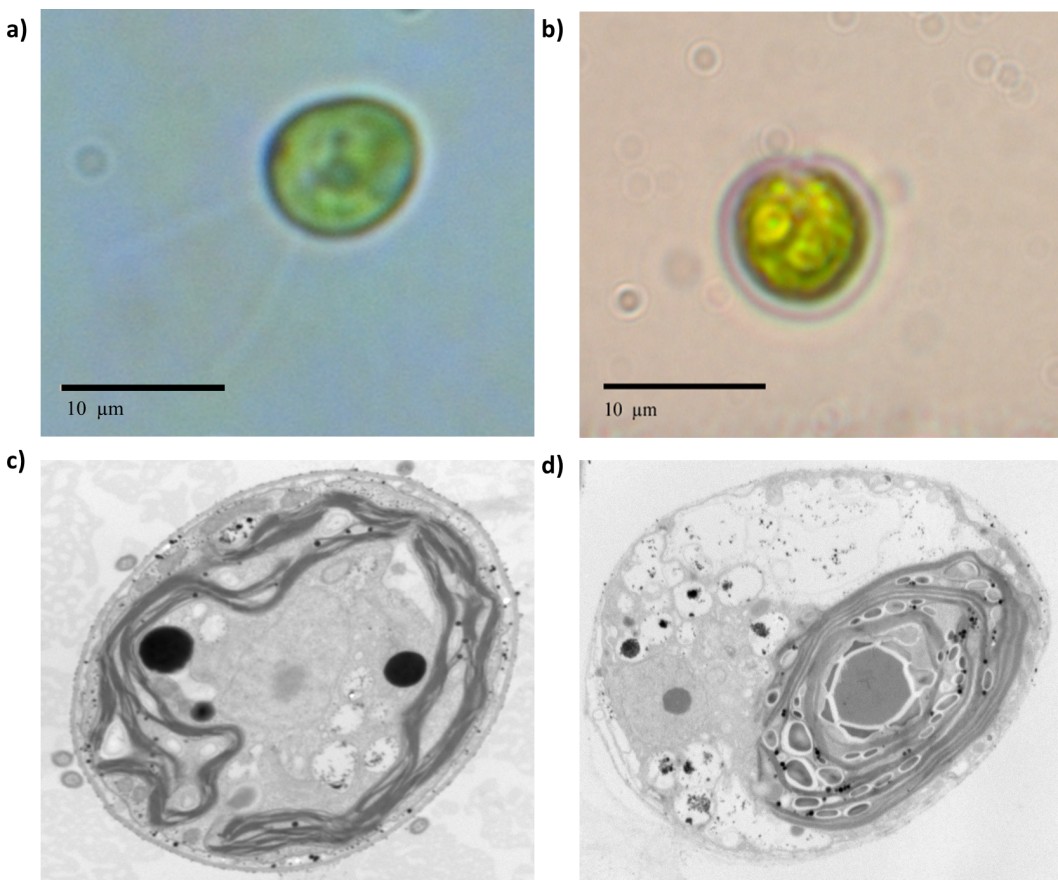

**Figure 2   Optic microscopy and TEM photographs taken at 12000× of wild-type and beryllium-resistant strains.** Size and ultrastructure differences were calculated. (A) and (C), wild-type strain optic and TEM photographs; (B) and (D), beryllium-resistant strain optic and TEM photographs.

and Be-resistant strains was approximately 0.7 and 0.6, respectively, showing close to 15% less quantum yield in the resistant strain than in the wild-type strain. A dose- and time-dependent decrease was observed in the wild-type strain. The Be-resistant strain showed stable values of $Y(II)$ at the different dose regimes with slight variations over time. However, the unexposed replicates of the Be-resistant strain (control replicates) showed a significant increase in $Y(II)$ at 72 h. This result can be explained by the cultivation of the Be-resistant strain in fresh culture medium without Be. A graphical representation of the $Y(II)$ values is provided in Fig. 3.

The extent of inhibition compared with control, as represented kinetically, indicated the time- and dose-dependent performance of wild-type and Be-resistant *Chlamydomonas* sp. strains (Fig. 3). All of the wild-type populations showed phytotoxicity at 24 h at all dose levels. The levels of inhibition in this strain increased with time, reaching as high as 40% at 72 h at 33 mg/l of Be. The inhibition curves for the Be-resistant strain showed negative values to a value of 0.05%, indicating that no inhibition occurred for 48 h. At 72 h, the inhibition values in the resistant strain increased up to 15% (Fig. 3). This result might

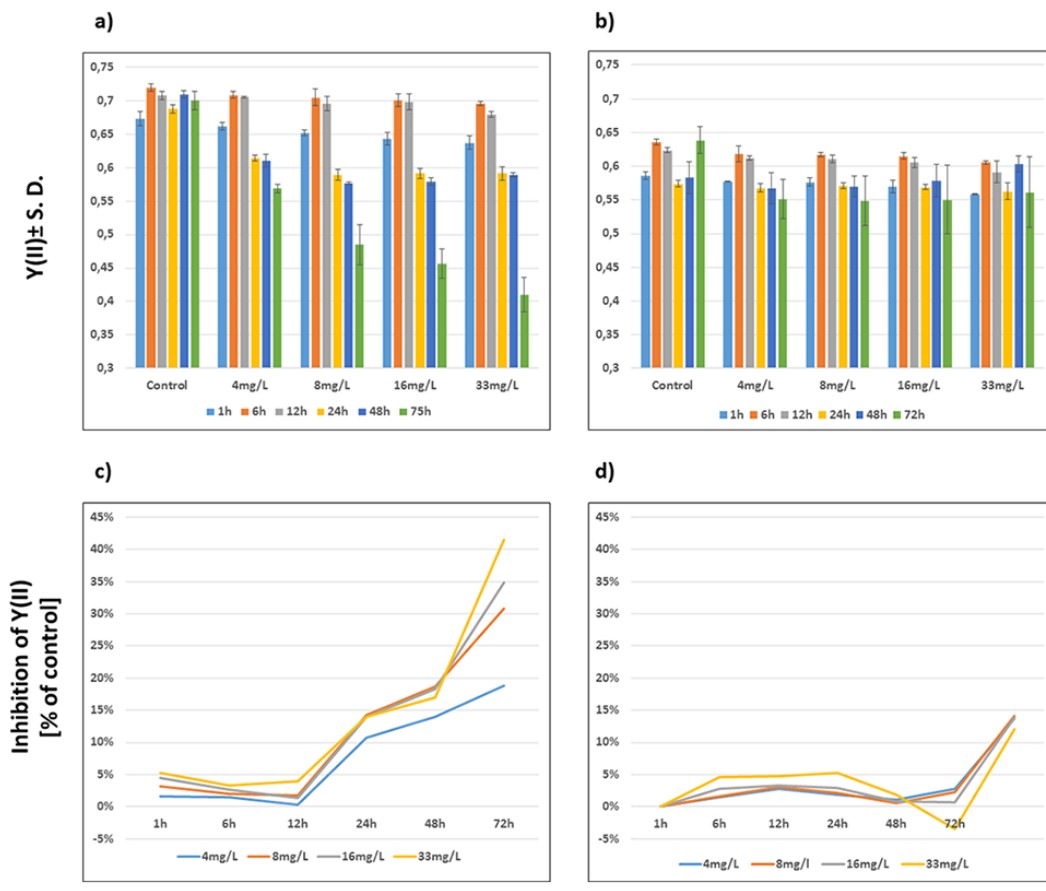

**Figure 3  Evolution of photosynthetic parameters.** Graphical representation of the effect of beryllium on $Y(II) \pm$ S.D (A, B) and % of $Y(II)$ inhibition compared to control (C, D) of wild-type (A, C) and Be-resistant (B, D) *Chlamydomonas* sp. strains. Measurements were obtained by using a modulated amplitude pulse fluorimeter (Imaging-PAM) at different beryllium concentrations (4, 8, 16, and 33 mg/l) and different times (1, 6, 12, 24, 48, and 72 h).

be the consequence of the increase in the *Y(II)* value of the control replicates due to the non-selective conditions of the medium.

## DISCUSSION

Although the dose–effect experiment resulted in population die-off at high doses of Be (more than 16 mg/l), the results of this study indicate that further incubations of wild-type *Chlamydomonas* sp. at high cell densities may restore the population, as indicated by the fluctuation analysis. By means of the fluctuation analysis, we studied the nature of adaptation to the Be selective pressure and found that the rescue variants arose during the propagation period prior to the exposition. The Be-resistant strain exhibited a low mutation rate and remarkable differences in morphological, photosynthetic and fitness characteristics from the wild-type *Chlamydomonas* sp. strain.

Over the past century, humankind has not only influenced natural selection but also tilted the balance toward Anthropocene selection. Therefore, we may have entered a new

human-dominated geological epoch (*Lewis & Maslin, 2015*). The crisis of biodiversity of the unseen majority in the earth's water mass is sometimes forgotten (*Whitman, Coleman & Wiebe, 1998*). This can destabilize aquatic ecosystems towards new ecosystems, particularly where population reductions are not severe. Under this new scenario, we can understand and predict phytoplankton drift. This basis of the aquatic trophic chain is becoming increasingly important in the fields of evolution and ecology.

The impact of pollution is often assessed by standardized toxicity studies (*Ventura et al., 2010*). We propose that further understanding of the evolutionary dynamics of populations and population ecology is needed. Several evolutionary approaches have been proposed to obtain reliable predictions, but evolutionary gradualism uncouples evolution from ecology (*Bell, 2013*). In contrast, predictive and evolutionary ecology attempt to implement a more reliable ERA by means of result integration and common pathway outcomes (*Straalen, 2003*; *Forbes et al., 2011*), and different pressure patterns influence the possible ecological results (*Baos et al., 2002*; *Tatters et al., 2013*). Via fluctuation analysis, we intend to analyze the adaptation capacity of a clonal population of *C. reinhardtii* under a maintained, strong selective pressure of Be dissolved in the environment.

Before performing the fluctuation experiment in the present study, we analyzed the dose effect of beryllium in *Chlamydomonas* sp. and determined the DL100 at which the demographic decline is sufficiently strong to promote extinction. Then, using fluctuation analysis, we studied the capability of the population to reverse the demographic threat. After three months, the populations showed recovery: Set 2 (control trials) exhibited population growth in all of the replicates and with lower variance between replicates than in Set 1, and Set 1 exhibited recovery in more than 90% of replicates and with higher variance than in Set 2. The rescuing cells were initially rare but increased in number under strong Be selection. The fluctuation analysis results indicate that the strain was capable of adaptation to Be. The initial genetic variability was limited to mutations that arose during the propagation period as the initial population was founded from a single cell. Survival under selective pressure involves a complex combination of phenotypic and behavioral changes (plasticity) without genetic modification (e.g., by the modification of gene expression (*Bradshaw & Hardwick, 1989*) and evolutionary genetic changes (such as mutations). Once plasticity is overcome by strong selection, the majority of the population fails to adapt; only mutations that confer resistance and subsequence selection can assure survival (*Lenski & Sniegowski, 1995*; *Sniegowski, 2005*).

We found that adaptation to high doses of Be occurred randomly via pre-selective mechanisms and not by means of specific adaptation that was induced by the presence of Be because if the latter had occurred, the final variation in Set 1 should have been very low. Therefore, in principle, plasticity and epigenetic effects did not feature in the adaptation process; it appears feasible that the adaptation was genetic. This is consistent with the Neo-Darwinian view regarding unfavorable selective environments: regardless of the presence or absence of the selective agent, the population presents rare, spontaneous, pre-selective mutations that confer resistance. Notwithstanding our experimental outcome and numerous other fluctuation studies in bacteria and phytoplankton (*Marvá et al., 2010*; *Marvá et al., 2014b*; *Carrera-Martinez et al., 2011*; *Romero-Lopez, Lopez-Rodas &*

*Costas, 2012*; *Costas et al., 2014*), several works suggest the existence of adaptive mutations in microorganisms (i.e., *Cairns, Overbaugh & Miller, 1988*; *Foster, 2000*). We are confident that in this case, adaptation took place randomly via selection for the preexisting mutant in the population. However, DNA sequencing and analysis are necessary to confirm this outcome, and the presence of post-selective mutations cannot be ruled out. The mutation rate obtained from the fluctuation analysis in the wild-type and Be-resistant *Chlamydomonas* sp. ($\mu = 9.61 \times 10^{-6}$) was high but in agreement with the typical mutation rates calculated for phytoplankton exposed to other metals (e.g., *García-Villada et al., 2004*; *López-Rodas, Maneiro & Costas, 2006*; *Sánchez-Fortún et al., 2009*; *Marvá et al., 2014b*).

The phenotypical, physiological, morphological, photosynthetic and fitness differences between the wild-type and Be-resistant *Chlamydomonas* sp. strains were substantial. The resistant strain substantially differed from the parental population with respect to the flocculated behavior in culture (without detectable movement of the flagella), microscopic shape, and the morphometry of subcellular structures. Morphometric analysis and stereology of cells indicated that the total cell volume of the Be-resistant strain is approximately 25% smaller than that of the wild-type strain. The relative volumes of chloroplasts and mitochondria were higher in the resistant strain but presented lower volumes of storage products, and no significant differences between the strains were found in the nucleus. Moreover, the coefficients of form indicate that the resistant cells present a more circular form. In addition, the fitness and the quantum yield of the Be-resistant *Chlamydomonas* sp. strain were significantly lower than those of the wild-type strain. Although form changes of the cell can be a response to environmental conditions (*Berg, Tymoczko & Stryer, 2002*), considering that the selective pressure was sufficiently strong to extirpate the population, the differences between the strains suggest that adaptation was mediated by large-effect, advantageous mutations that substantially contribute to evolutionary change (*Eyre-Walker, 2006*).

Due to the reduced fitness value of the Be-resistant *Chlamydomonas* sp. and the observed mutation rate, it is likely that mutations arose continuously with each generation and were subsequently eliminated by genetic drift or natural selection. However, as the strength of the selective pressure that favors the advantageous mutation increases, the probability that the population presents a substitution increases (*Eyre-Walker & Keightley, 2007*). Advantageous mutations are rare (*Eyre-Walker & Keightley, 2007*; *Orr, 2010*). In the present study, the mutation-selection balance resulted in 10.42 mutant cells per million wild-type cells; in natural environments, wild-type *Chlamydomonas* sp. populations are sufficiently large to ensure that a sufficient number (i.e., around ten per million) of Be-resistant cells would survive a Be spill. Although the initial variability and plastic response likely have greater importance under multifactorial natural scenarios, studying genetic adaptation can lead to the identification of genetic outcome pathways and achieve better ERA.

New concepts that integrate both ecological and evolutionary approaches, such as evolutionary rescue (ER), are arising that attempt to explain the outcomes for natural populations under rapid environmental changes (*Gonzalez et al., 2013*). ER involves changes in evolutionary dynamics over short time scales, absolute fitnesses and a strong effect of genetic variants (*Bell & Gonzalez, 2009*). Although phytoplankton are ideal

for studying the ER approximation, our approach excludes the effect of competition. Specifically, we employed fluctuation analysis to study the population's response under an environmental change that was sufficiently severe to produce a demographic reduction over a sufficiently short time scale to promote extirpation. Moreover, the demographic factor (population size) had a marked influence on the results of the experiment, affecting the initial genetic variability and enabling the adaption processes to be distinguished between pre-existing genetic variability and post-selective *de novo* mutations. *Orr & Unckless (2014)* described a theoretical model to identify the most likely genetic factor of ER, i.e., new post-selective mutations or pre-existing variability. The model determined that the mutation frequency for the population ($q$) should be higher than the mutation rate ($\mu$) divided by the wild-type fitness under selective pressure ($m^{wBe}$), suggesting that standing genetic variation is more likely ($q > \mu/m^{wBe}$). Considering that fluctuation analysis exercised extreme selection, $m^{wBe}$ was near zero (the dose–effect curve showed no growth); when $q$ is higher than $\mu$, the adaptation is more likely to be pre-selective. Perhaps the strong selective pressure did not allow the time necessary for the appearance of post-selective adaptive mutations.

Future studies on this topic could include the use of specific biosensors to detect Be based on fluorescence measurements or bioaccumulation by the algae biomass. The high cost of detecting and removing Be from water can be alleviated with low-cost algae technologies. As Be is present in the aquatic environment at orders of magnitude of micrograms per liter, avoiding the risk of harmful effects to animals and humans is necessary. The chemistry of the bedrock, water acidity and acidification influence Be mobilization (*Edmunds & Trafford, 1993*; *Neal, 2003*), especially in severely polluted and acidified basins, lixiviation mining ponds, acid mine drainages and industrial discharge. Due to beryllium's atomic number ($z = 4$), an analysis of the bio-accumulation within the beryllium-resistant cells could not be performed with X-ray energy dispersive spectroscopy (XEDS) (the detection limit of the technique is determined by the atomic number of the element, more than $z = 4$). However, we do not rule out the possibility of beryllium recovery from aqueous environments using the Be-resistant *Chlamydomonas sp.* strain obtained from the fluctuation analysis, as the biological recovery of Be by freshwater and marine microalgae has been described (*Gatewood & Sneddon, 1990*; *Ishikawa, Kagaya & Saga, 2004*).

## CONCLUSION

We performed a fluctuation analysis experiment to simulate a scenario of a large discharge of Be in freshwater (such as may occur in a lixiviation mining pond or in an industrial spill) in which the toxicity exerts selective pressure, resulting in catastrophic effects on phytoplankton and damage to the ecosystem. Under this scenario simulation, an adaptation phenomenon occurred and suggested that under appropriate ecological conditions, *Chlamydomonas* sp. could evolutionarily adapt sufficiently rapidly to such pressure as to prevent extinction. The mutation rate of this microalgae is very low (approximately $10^{-6}$), but given a frequency of approximately ten adaptive cells per million wild-type cells, population survival can be assured. The adaptive phenotype differs substantially

from the wild-type genotype. Considering that the population propagated from a single cell (the genetic variability present in the population appeared by means of mutation during replication) and that the phenotypic plasticity response is not consistent with the fluctuation analysis outcome, adaptation most likely occurred via advantageous mutations. However, DNA sequencing is necessary to confirm this hypothesis. *Chlamydomonas* sp. was able to successfully adapt to 33 mg/l of beryllium.

## ACKNOWLEDGEMENTS

Prof. Antonio Flores Moya and Prof. Victoria López Rodas for their invaluable advise. To Lara de Miguel Fernández for her excellent technical support.

### Funding

This work was supported by the Spanish Secretaría de Estado, Investigación, Desarrollo e Innovación (Supported Grants; CTM-2012-34757 and CTM-2013-44366-R). The funders had no role in study design, data collection and analysis, decision to publish, or preparation of the manuscript.

### Grant Disclosures

The following grant information was disclosed by the authors:
Spanish Secretaría de Estado, Investigación, Desarrollo e Innovación: CTM-2012-34757, CTM-2013-44366-R.

### Competing Interests

Estéfano Bustillo-Avendaño and Camino García-Balboa are employees of I+D+I, Biotechnological Environmental Solutions S.L., Madrid, Spain.

### Author Contributions

- Beatriz Baselga-Cervera conceived and designed the experiments, performed the experiments, analyzed the data, contributed reagents/materials/analysis tools, wrote the paper, prepared figures and/or tables, reviewed drafts of the paper.
- Eduardo Costas analyzed the data, contributed reagents/materials/analysis tools, wrote the paper, reviewed drafts of the paper, statistics.
- Estéfano Bustillo-Avendaño performed the experiments, contributed reagents/materials/analysis tools, prepared figures and/or tables, reviewed drafts of the paper.
- Camino García-Balboa analyzed the data, contributed reagents/materials/analysis tools, wrote the paper, reviewed drafts of the paper.

### Data Availability

   The raw data has been supplied as Supplemental Information.

## Supplemental Information

Supplemental information for this article can be found online at http://dx.doi.org/10.7717/peerj.1823#supplemental-information.

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
