# Peer review of "Adaptation prevents the extinction of Chlamydomonas reinhardtii under toxic beryllium"

_PeerJ, doi:10.7717/peerj.1823_

## Round 0.1 · original submission · Major Revisions

The reviewers are very positive about your work, but they generally agree that the text suffers from lack of clarity at several key points. Please address their points carefully. Pay special attention to the statistical analysis suggestions.

A few personal comments:

the culture of exponentially-growing cells (at a concentration of 1.5 * 10^5 cells/mL) dies completely with the highest-dose of Be, and no growth is observed even after 28 days. However, in set 2 (apparently at the same Be concentration and cell density) growth was observed in all 47 replicates. I believe this is simply a matter of a longer incubation time, but you may want to highlight and explain this apparent discrepancy.

fig S1 is not very clear.... Could you please explain how to interpret it ?

line 193 "in which 11. 68 generations took place" . Do you mean "in which an average of 11.68 generations took place" ?

line 193 : "population density" lacks volume units. Do you mean 10^5 cells/mL ?

line 196 seems to suggest that Set 2 was innoculated at the end of growth of Set 1, but does not tell the source of the cells: were thay all taken from a single tube of set 1, or were they taken from a 97th tube, inoculated at the same time as set 1 was innoculated? Please rephrase, or provide a clear graphical timeline

lines 359 and 362 : you have "p< using" and "(p)". Shouldn't it be "p<(some number)" and "p=(some number)" ?

Reviewer 1 ·

Basic reporting

English was not clear in lots of places. e.g. lines 82-83, 103-106, 167-169, 310-312, 335-342, 350-357, 424-443, 542-544, 547-553.

Introduction was sufficient (however see below for areas that need improvement).

Structure is OK.

Figures are fine. However, Fig S2 should be included in the main text with a well explained legend. Also a descriptive figure is direly needed for the fluctuation experiment setup.

Experimental design

The idea behind the work is clear. It took me a while to understand what the authors meant in lines 183-274. English needs a lot of work as detailed below. A descriptive illustration of the experimental setup of the fluctuation experiment will make things clear for sure. Additional information is needed when it comes to equations calculating the different rates and coefficients (as explained below). I personally believe that the authors are not doing themselves a favor. The materials and methods section is not doing them justice. It appears that they have done a lot of work but are selling themselves short with the unclear description of methods in some places.
Statistics is good and sufficient.

Validity of the findings

While data in Tables 1 and 2 are sufficient for a Table, honestly the explanation in text is terrible. It took me a while (about an hour or so) to understand what they meant in lots of places (especially lines 333-357, and 424-443).
Conclusions are well supported.

Additional comments

Throughout the text change mg/l-1 to either mg/l or mg.l-1
Throughout the text change exposition to exposure.
Fig S2 should be in the main paper (not supplementary). However if so it needs a very good legend. The figure is showing too many variables (time, dose, as well as the yield or % inhibition) so care should be taken to make this as clear as possible.
Add a figure to illustrate the flucuation experiment. It was not clear (took me a while to understand the differences between the two sets). In the figure make it clear that set 2 cultures are inoculated from the end of the growth phase in set 1. After I figured this out, it made things easier to understand.
Line 40: change an to a
line 52: led to
line 57: spell out ER
line 61: change have to has
lines 82-83: not clear. rewrite
line 98: approaches
Lines 104-106: not clear. rewrite
Last paragraph of the introduction should summarize the study. remove the senetnce "We emphasize the importance of the study of toxicological behavior" and instead add In this study, the population's evolutionary .....with dissolved Be was first evaluated to determine the level of tolerance of the wild type". Also at the end of this paragraph add a sentence that summarizes the results.
Line 164: change to: ). The beryllium salt was ....
Lines 167-169 were not clear
In lines 227 and 229 add something like this: a high q (> than x) means .... while a low q(< than y) means ....
Similarly for s add a sentence describing what high and low values mean.
Lines 238-239. Please briefly describe the stereological and morphometric methods and what they produce. I did get eventually what you mean when I read this section in the results but it was definitely not clear to me in the materials and methods.
Line 262: shouldn't cell value be cell volume.
Line 288: change imagines to images
Lines 307-316: I wasn't sure what was being measured here. You need to rewrite the sentence that starts in line310 and ends in line 312. I think you meant that the measurement was taken immediately before and right after the 800 msec of SP. Change F in line 315 to Ft. Spell out PS. Again add a sentence that describes what a high and low quantum yield would be.
Line 321: add a sentence that values below x% are considered phytotoxic

Results section: needs najor rewriting especially lines 333-357 and 424-443.
Table 1. change the title to % decrease in viability. The results here are not well explained. If I got it, the numbers in the table are the % dead cells pretty much compared to the control. why would this number go down then up then back to 0 when exposed to 0.8 mg/l Be? why was the effect not similar with 4-16 mg/l. Why not try a conc in between 0.8 and 4 mg/l. Please explain this better.
table 2: Now I get what the numbers mean but it took me a while. the explanation text for the table need to be rewritten.
Figure S2 needs to be added to main table. I still dont get what the authors want to say in lines 424-443. Please rewrite this section for clarity and make good references to the sub figures A-D.

Discussion:
First paragraph in discussion should be the major findings in the study.
Line 511: I thought the relative volume of nucleus did not change significantly according to Table 3.
Lines 547-553: interesting. Did you measure mwBe. I dont think you reported this value in text. Do your data apply to this model. Please explain clearly.
Line 559: what does many orders of magnitude of microgram per liter mean?
Sentence in lines 566-569 is not clear.
Line 587: italicize Chlamydomonas.

Reviewer 2 ·

Basic reporting

The manuscript is generally readable and well structured, but there are some grammatical and spelling mistakes that need correcting.

The Introduction (esp. line 82) appears to indicate that the ability of microbes able to survive and adapt to pollutants is a new phenomenon, but of course many microorganisms, particularly extremophile microorganisms, have long been known to have a substantial ability to tolerate and quickly adapt to excessive stress conditions in their environments (as indeed the authors have discussed in many of their previous publications). This should be clarified in the Introduction.

Scale bars need to be present on the images in Figure 1.

Experimental design

The authors of this manuscript have examined the responses and adaptation of the freshwater microalga Chlamydomonas reinhardtii to Beryllium (Be), with the aim of understanding adaptive responses and survival to acute abiotic stress, in order to predict future biodiversity trends.

From previous studies, is Be known to highly accumulate within the cell? Is anything known about the mechanisms of Be toxicity at the cellular level? Have the authors quantified Be surface binding and internalisation? It would have been useful to correlate toxicity responses with Be sorption. Although the authors state that Be accumulation cannot be performed by XEDS, previous studies have quantified Be uptake by ICP-MS (e.g. Ding et al 2009 Metallomics 1: 471-478).

Previously used, well validated methods are used to determine pre-selective or post-selective adaptation, and to determine population dynamics. Physiological characterization of the cells appears to have been well performed.

In addition to determining PSII quantum yield, it might be useful to determine other photosynthetic parameters from the chlorophyll fluorescence data such as Fv/Fm ratio values.

The Be-resistant cell appears to show increased starch granule within the chloroplast – a change in carbon storage characteristic is often seen as a stress response in such cells. Was this seen consistently in the Be-treated cells? If so the authors may wish to comment on this observation.

Validity of the findings

The data appears to be statistically valid and the conclusions are appropriate.

With regard to the data in Table 1, can the authors confirm that the cultures for this experiment were only inoculated once, at the start of the growth period, and exactly what does 100% inhibition of the cell number mean? For the 16 mg/L Be treatment, there was 100% inhibition of cell number after 24 h - so there were no live cells? But by day 14 there was some cell abundance. Can the authors clarify this point?

Flocculation is mentioned in passing, but can this be quantified?

·

Basic reporting

The manuscript meets the general standards required of PeerJ in terms of language, structure, and figures. I think a revised version of the manuscript could be stronger if it there was a more focused development of the introduction. For example, in the Introduction, significant space is devoted to biodiversity declines, the importance of phytoplankton, and environmental risk assessment. Because beryllium toxicity is not something that everyone thinks about, I thought the fourth paragraph was useful. But it isn't until the fifth paragraph, the authors introduce the idea of experimental evolution. Less detail is devoted to this, even though this is perhaps the most interesting part of the manuscript, at least in my opinion.

The arrangement of the major sections seems to make sense. That is, there's the section on fluctuation analysis and then there's the second part that focuses on the characterization of the mutants. I would say, however, that this structure could be more explicit. For example, the text relating to characterization of mutants was very long relative to the sections on fluctuation analysis, mutation rate, and selection. It might help the flow of the manuscript if the two major sections were more balanced.

There is a first mention of evolutionary rescue in the abstract, but only by abbreviation. It then appears late in the discussion. Seems like it could be appropriate to talk about, but needs better development and introduction.

There are many typographical and grammatical errors throughout the paper. Combined with some of the organizational issues mentioned above, this greatly diminishes the potential of this manuscript. If this manuscript is to revised, careful attention should be paid to improving these parts of the paper.

Experimental design

The experimental design is reasonable. I'm not an expert on fluctuation analysis, but from a quick search, it seems that there are some rather sophisticated ways of analyzing data from such experiments, including maximum likelihood analysis. The description of the quantitative analysis of the fluctuation analysis was not very elaborate; I think this could be improved upon in revision. Why is there such a difference in sample size between "Set 1" and "Set 2" (i.e., 87 vs. 47)? Does this unbalanced design effect the fluctuation analysis in any way?

In general, more attention could be paid to the statistical analyses. For example, on line 325, the statistical analysis section is very short and only mentions treatment of the ultrastructural data. What about other parts of the study? For example, is there a way to more succinctly summarize and synthesize data from Table 1? The current manuscripts seems to treat these data in a rather qualitative way with no accompanying statistics that i could find.

Validity of the findings

I don't have any major concerns about the validity of the findings outside of what might be learned giving comments regarding the statistical analyses expressed in the Experimental Design part of this review.

Additional comments

Line 45 -- "We propose that fluctuation analysis is a good empirical procedure to study adaptation under strong selective pressures." As the authors state, this method has been around for a long time, so the statement seems strange as it suggest a novel approach.

Why was Chlamydomonas was chosen? is there any record of it having been exposed to high levels of Be? I would think that it would strengthen the claims of relevance of this study if the strain chosen was from an area at-risk of Be pollution (ex. an algae strain from a body of water near a mining site).

The use of the word "drift" implies or could be confused as genetic drift, which does not seem to be a focus of the manuscript.

L86: Phytoplankton "are"

L108: Perhaps there is too much detail on the industrial uses of Be and it's physical properties?

L116: what is the mechanisms of toxicity?

L135: "halfway house" is strange; has other meanings: https://en.wikipedia.org/wiki/Halfway_house

L217: "Population dynamics" doesn't seem to be the best description of the data and experiments for this study.

L222: I don't understand the use of the word "effective" in this sentence.

L457-459: The line starts with "Without diminishing the significance of this type of study." I'm not sure how this sentence connects to the rest of the section. At this point I didn't think that suggesting future work would diminish the significance of this paper.

L 339 and elsewhere: I think "exposure" instead of "exposition"

L350: what does cell "destruction" mean?

L476: an "a" should be added before "mutations"

L523: I think the "be" in this sentence should be changed. I get the impression that the sentence is set up to bring up the number of expected mutations.

L585-585: I'm not sure if this sentence is referring to what you predicted going into the experiment or what the data suggests happened.

---

## Round 0.2 · Minor Revisions

No further changes to the science present in the paper are needed. However, you should polish the grammar and language quality, preferably with the help of a native English speaker.

Reviewer 1 ·

Basic reporting

The article improved tremendously from the first version. I don't have any concerns except that overall the English need to be revisited. Sentence structure and grammar both need a lot of improvement.
Figures are well explained in legends. Figure 1 addition was a great help.
Tables are also well explained.

Experimental design

No comments to report here.

Validity of the findings

Data is statistically sound and are made available.
No other comments

·

Basic reporting

NA

Experimental design

NA

Validity of the findings

NA

Additional comments

The revised manuscript is much improved and I think the authors did a good job of addressing the concerns expressed in my original review. However, the manuscript needs to be edited again for grammar and typographical errors.

---

## Round 0.3 · accepted · Accept

All requested revisions have been performed.